# Numerical Investigation on the Correspondence between the Damping and Coefficient of Restitution (COR) in Rockfall Movement

**Yan Ai** **, Hongyan Liu * and Ziwei Ge**

School of Engineering & Technology, China University of Geosciences (Beijing), Beijing 100083, China
* Correspondence: lhy1204@cugb.edu.cn

**Abstract:** The rockfall process is characterized by bounces of a block on the ground. The coefficient of restitution (COR), which indicates the degree of rockfall energy dissipation, has a significant effect on the rockfall trajectory. The 3-dimensional Distinct Element Code (3DEC) is an effective tool to study the rockfall trajectory, and the damping can reflect the COR in numerical modeling. However, the relationship between damping and COR is not understood. A field test is numerically modelled to investigate the correspondence between damping and COR. A series of damping–COR correspondences are obtained and compared with the field test and its previous numerical simulation to verify the rationality of the correspondences. Then, the damping–COR correspondence is adopted in a typical rockslide in Yunnan province, China. The numerical results show that the proposed method is in good agreement with practical engineering. This study provides a new method for predicting rockfall trajectory.

**Keywords:** rockfall; distinct element method; coefficient of restitution; damping; movement characteristics; polyhedral blocks

## 1. Introduction

Rockfalls are common geological hazards in mountainous areas that are characterized by rapid, sudden, and highly destructive conditions that often threaten the safety of nearby infrastructure (such as bridges, roads, and railways) and residents [1]. Therefore, it is necessary to evaluate the occurrence probability of rockfall hazards and predict their trajectories when designing protective structures. Research on rockfall movement characteristics has mainly focused on the movement trajectory, kinetic energy (and its evolution), and block stop position [2]. The coefficient-of-restitution (COR), trajectory, velocity, and kinetic energy are key parameters used to characterize rockfall movements, and they are directly considered in the design of protective countermeasures [3,4]. These parameters can provide a basis for selecting protective structures and their optimal location, size, and strength [5]. Due to the scale, the propagation process of rockfalls is dominated by different physical processes. Fragmental rockfalls are often analyzed as the movement of a single rigid body, which moves under the action of gravity and discontinuously interacts with the substrate [6]. However, the falling of rock mass is usually regarded as a particle flow with a lumped mass to maintain fairly continuous contact with the bedrock for dynamic analysis [7–10]. The boundary between fragmental rockfalls and rock mass fall is transitional. The propagation process of rockfalls is usually hybrid-analyzed [11]. Therefore, the study of rockfall movement characteristics is particularly important for rockfall risk assessment, prediction, and prevention [6].

Many methods, such as theoretical calculations [6,12], laboratory experiments [13–15], field experiments [16–18] and numerical simulations [8,19,20], have been adopted to study the movement characteristics of rockfalls. The jump in the trajectory of the falling rock is only due to the irregular shape of the block and its rotation. In complex rockfall motion

simulations, COR is mainly used to describe trajectory rebounds, such as movement COR ($R_v$), tangential COR ($R_t$), and normal COR ($R_n$). These parameters are especially important because they reflect the energy dissipation throughout the impact process and affect the accuracy of rockfall trajectory prediction [3]. However, due to the above reasons, it is still difficult to accurately calculate the trajectory of a rockfall. Therefore, it is necessary to comprehensively study the dynamic rockfall process from the initial instability to movement until stopping.

Due to the complex contact transformations among blocks of various shapes and slopes in complex terrain [11], a suitable discontinuous dynamic numerical method is needed to analyze the rockfall movement process. Many discontinuous numerical methods can be adopted to simulate the movement process of rockfalls, such as the three-dimensional discrete element method (3D-DEM) [9,21], three-dimensional discontinuous deformation analysis (3D-DDA) [11,19], and numerical-manifold (NMM) [22], general particle dynamics (GPD) [23], peridynamics (PD) [24], and nonsmooth multibody systems dynamics methods [25]. Theoretically, the DEM uses an explicit time integration approach, but DDA and the NMM use implicit time integration. The unnecessary global matrix generation in each calculation step in explicit time integration increases the calculation speed in DEM relative to DDA and NMM, especially when there are numerous blocks [9].

The mechanical behavior of energy dissipation is involved when simulating discontinuous problems such as landslides and rockfalls with DEM. As an approach to alleviate unnecessary vibration caused in a dynamic system, damping is used to characterize the energy dissipation in DEM simulation. Researchers have conducted extensive research on the damping of DEM. Jiang et al. proposed that there is less quantitative research on damping in the form of energy dissipation in numerical methods [26]. Therefore, it is necessary to introduce a suitable damping model into the numerical method to more accurately simulate the energy dissipation of the block system [27–29]. Then, mass damping, viscoelastic damping, adaptive damping, and Rayleigh damping are embedded into numerical methods to describe the energy dissipation of blocks [25,30–33]. Ge et al. showed that damping affected movement, but did not quantify the effect [34]. Jiang et al. found that viscous damping was suitable for simulating quasistatic dynamic processes [26], while Rayleigh damping is often used to analyze dynamic processes [35].

Despite damping having been widely used in DEM simulations, damping parameter selection is not understood well. The rationality of damping parameter selection was usually verified by comparing numerical results with historical disasters or field tests. Wu et al. applied a local damping value of 0.8 to landslide simulations with 3DEC and obtained results that agreed with the actual affected area [9]. Wu et al. used the 3DEC model without damping to simulate a landslide case, which was also verified in the actual affected area [8]. Xu et al. applied global damping value of 0.02 to a particle DEM-based landslide model and validated the results through field investigations [10]. These studies showed that the value of damping focuses on verifying historical data and field tests, but the values are divergent and lack systematic quantitative research. Moreover, the value of damping for numerical simulations in areas lacking historical data and field tests remains to be studied, and the correspondence between damping and COR also needs to be studied in detail.

This study numerically models a field test to investigate the correlation between COR and damping parameters. A series of damping–COR correspondences are obtained and compared with the field test and its previous numerical simulation to verify the rationality of the correspondences. The obtained corresponding values were applied to the analysis of the practical rockfall engineering on the basis of the COR of similar geological conditions.

## 2. Numerical Investigation on the Correlation of the COR and Damping Parameter

### 2.1. Brief Introduction to 3DEC

In this study, a DEM code called 3DEC was adopted to simulate the process of rockfall movement. The 3DEC code [35] enables the numerical simulation of contacts between polyhedral and planar boundaries. In a discontinuous numerical model, the following two

behaviors in discontinuous systems must be considered: (1) the behavior of blocks and (2) the behavior of discontinuities.

Figure 1 shows the basic computational principle of 3DEC. The force/displacement law is applied to identify all contact forces from known displacements. The equation of movement in Figure 1 provides grid-point accelerations, $\ddot{u}_i$, in the X, Y, and Z directions, which resulted from the summation of the forces built up at each grid point using Equations (1) and (2). The accelerations were numerically integrated to determine the velocities and integrated again to determine the displacements. The next time step was initiated again using this new set of displacements [9].

$$\ddot{u}_i = \frac{\int_s \sigma_{ij} n_j ds + F_i}{m} + g_i \tag{1}$$

$$F_i = F_i^c + F_i^l \tag{2}$$

where $\sigma_{ij}$ is the zone stress tensor; $s$ is the surface enclosing the mass, $m$, present at the grid point; $n_j$ is the unit normal to $s$; $F_i$ is the resultant of all external forces applied to the grid point; and $g_i$ is the gravitational acceleration. $F_i^l$ presents the summation of the external applied load, and $F_i^c$ is the result of the subcontact forces that exists only for the grid point along the block point [9].

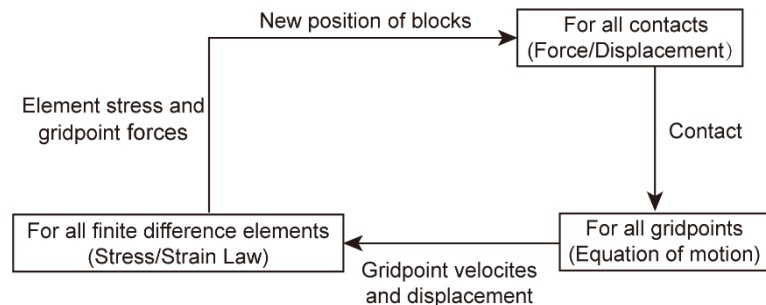

**Figure 1.** Calculation cycle of a DEM method with deformable blocks [35].

To simulate discontinuous behavior, the contact between these two blocks must be calculated (Figure 2). The common plane (c–p) was used to judge and compute contacts between blocks [36]. c–p was located between the centroids of these two blocks, with the unit normal vector pointing from one centroid to the other.

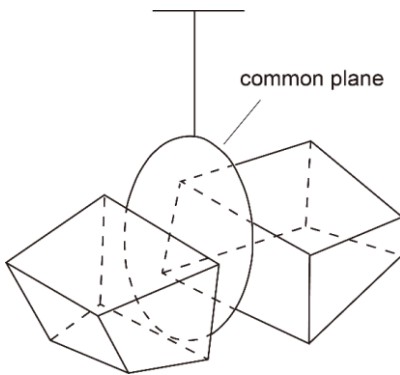

**Figure 2.** Sketch of the common plane in a 3D DEM [36].

When block contact occurs in 3DEC, normal joint stiffness $k_n$ and normal contact damper $\eta_n$ are connected in parallel in the normal direction (Figure 3a). Furthermore, in the shear direction (Figure 3b), shear joint stiffness $k_s$ and shear contact damper $\eta_s$ are connected in parallel and connected to the Mohr–Coulomb slider. During calculations,

contact dampers are used to mitigate unwanted contact vibrations. Damping is partly due to energy loss as a result of slippage along the contacts of blocks within the system, internal friction loss in the intact material, and any resistance caused by air or fluids surrounding the structure [35]. Damping can be divided into global and local. Global damping is used in the numerical integral calculation of velocity, and local damping is used in the calculation of contact forces. Both the joint stiffness and the contact damper are numerical parameters. Cundall [36] and Hart [37] discussed the detailed theory and computational algorithms.

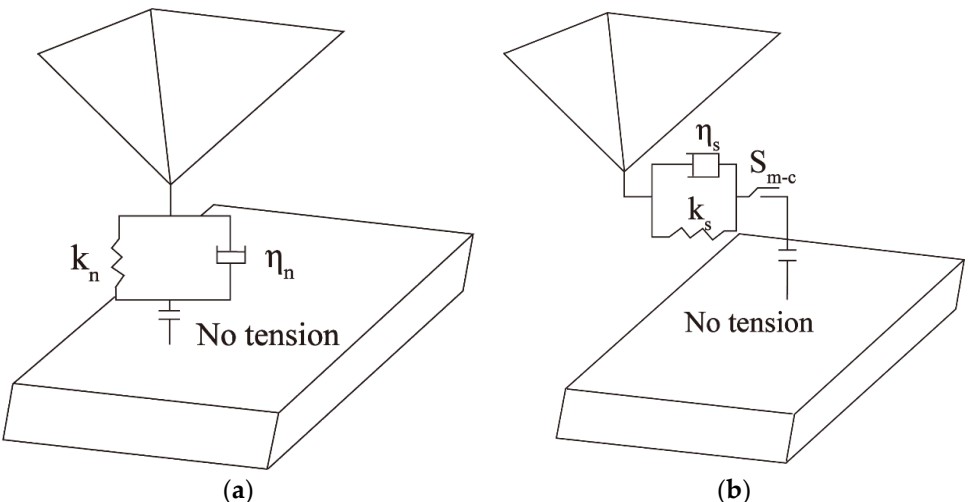

**Figure 3.** Sketch of the mechanical behavior of contacts in a 3D DEM. (**a**) Normal contact; (**b**) tangential contact.

Two types of damping (mass-proportional and stiffness-proportional) are available in 3DEC. Mass-proportional damping applies proportional force to absolute velocity and mass, but in the opposite direction to that of velocity. Stiffness-proportional damping applies proportional force to the incremental stiffness matrix multiplied by relative velocities or strain rates to contacts or stresses in zones. In 3DEC, the two forms of damping may be used separately or in combination. The use of both forms of damping in combination is termed Rayleigh damping, whose matrix $C$ (Equation (3)) consists of proportional components to the mass ($M$) and stiffness ($K$) matrices [35]:

$$C = \alpha M + \beta K \tag{3}$$

where $\alpha$ is the mass-proportional damping constant and $\beta$ is the stiffness-proportional damping constant.

The input parameters required to specify Rayleigh damping in 3DEC are $f_{\min}$ (input parameter $f_{\text{req}}$) and $\xi_{\min}$ (input parameter $f_{\text{crit}}$). Dynamic analysis is typically performed to obtain the frequency-independent damping $\omega_{\min}$ of materials at the correct level, as given by Equation (4):

$$f_{\min} = \omega_{\min}/2\pi \tag{4}$$

Since damping in geologic media is predominantly hysteretic and hence independent of frequency, $\omega_{\min}$ is usually chosen to lie in the center of the range of frequencies present in the numerical simulation [35]. In this way, hysteretic damping is approximately simulated. The 3DEC manual states that the value of $\omega_{\min}$ is 2–5% for geological materials, and 2–10% for structural systems [35]. After $f_{\min}$ is determined, the value of $\xi_{\min}$ is adjusted according to the actual site conditions until the model results conform to the practical test data. The predominant frequencies of Rayleigh damping are neither the input frequencies nor the natural modes of the system, but a combination of both [35].

The 3DEC manual states that mass-proportional damping is generally used to solve quasistatic problems using finite difference schemes. However, it is improper to consider

any mass damping for some problems involving large block movements because the block movement might be artificially restricted. Examples of such problems include any problems involving the free flow or fall of blocks under gravity and the impulsive loading of blocks due to explosions. In such cases, it may be appropriate to use only stiffness-proportional damping [35].

### 2.2. Verification of the Numerical Model

A field test [38] was numerically modelled to investigate the correspondence between damping and COR, and its numerical simulation [20] was used to for comparison.

In this field test (Figure 4a), nine selected cylindrical blocks with almost identical shapes and masses, a density of 2630 kg/m$^3$, an average diameter of 14 cm, and an average length of approximately 11 cm were considered. A slope with uneven surfaces and three plateaus s selected. The slope surface is composed of exposed weathered mudstones and accumulated gravel layers. The nine blocks are freely released one by one under same conditions with only slightly random variations in initial orientation and position at the top of the slope. The rockfall trajectories, horizontal run-out distances, lateral offsets, and arrival time were recorded for nine single runs of each block in the test [20,38]. Figure 4a shows the randomness of nine rockfall trajectories under nearly identical release conditions.

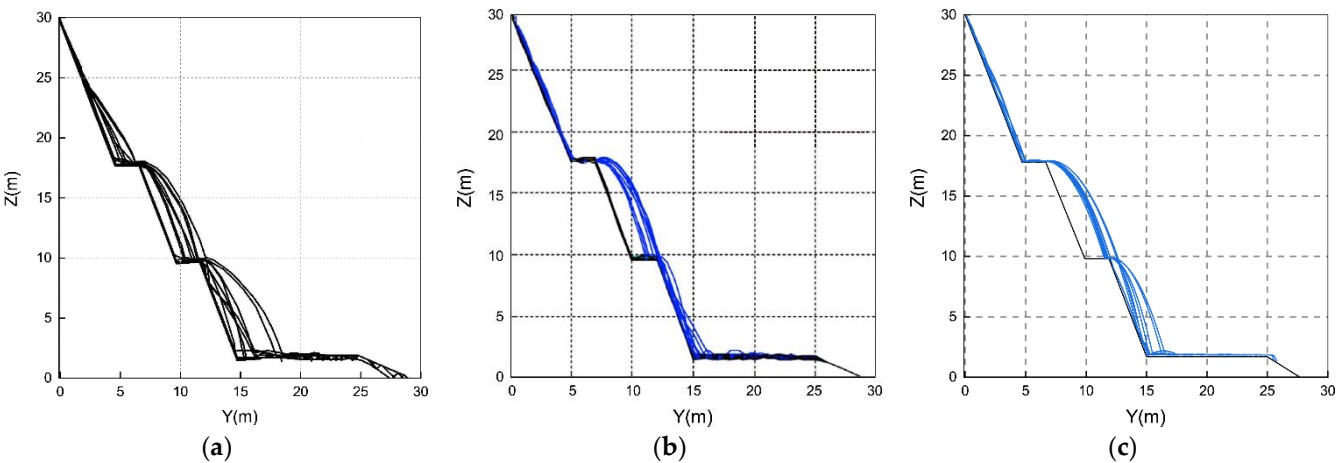

**Figure 4.** Comparisons of trajectories between test and numerical predictions. (**a**) Test observation [38]; (**b**) previously numerical prediction [20]; (**c**) numerical prediction in this study.

Cylindrical blocks and flat slopes were generated in numerical simulations on the basis of the actual size and shape of blocks determined from field tests. According to the existing case [9], the normal and shear joint stiffnesses of the rockfall and slope were set to be $k_n = 4.5 \times 10^6$ Pa/m and $k_s = 3.0 \times 10^6$ Pa/m, respectively, on the basis of trial and error to match the ratio of $k_n/k_s = 2/3$ suggested by Cundall and Strack [21]. The friction angle of the block was set to 35° according to a field test [38]. When the optimal contact stiffness was obtained, there was no block penetration or abnormal block movement during the rockfall simulation.

The 3DEC manual states that problems involving free fall and the bouncing of blocks on fixed bases require accurate modeling using the COR [35]. The COR is typically used to define the velocity change before and after a collision in a rockfall movement (Figure 5):

$$R_n = V'_n / V_n \tag{5}$$

$$R_t = V'_t / V_t \tag{6}$$

where $V_n$ and $V'_n$ are the normal components of the block velocity before and after a collision, respectively. $V_t$ and $V'_t$ are the tangential components of the block velocity before

and after a collision, respectively. Table 1 lists the COR's field test results and theoretical calculations for various slope materials.

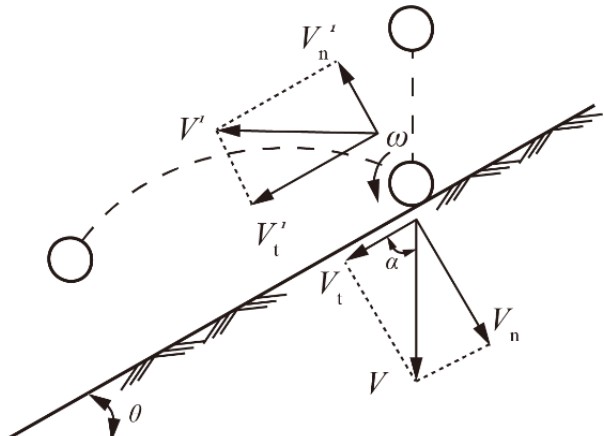

**Figure 5.** Schematic diagram of the COR calculation.

Considering the rock mass case detailed in subsequent chapters, a typical COR $R_n = 0.35$ was selected as the simulation benchmark. As shown in Figure 5, a falling rock is released at a certain height above the slope to allow it to fall freely. Only the velocity in the first collision between the rockfall and the slope are considered to prevent the influence of rockfall rotation on the velocity component. Then, the normal and tangential CORs along the slope are calculated from the velocity component. After repeated tests, when the Rayleigh damping parameters $f_{min} = 0.015$ and $\xi_{min} = 0.95$, the normal COR $R_n = 0.35$ and the tangential COR $R_t = 0.521$ are obtained, which conform to the principles of rockfall movement.

**Table 1.** COR values of different slope materials in field tests and theoretical calculations.

| Author(s) | $R_n$ | $R_t$ | Slope Description |
|---|---|---|---|
| Wang et al. [39] | 0.31 | 0.71 | Talus |
| Hoek [40] | 0.32 | 0.82 | Talus cover |
| Dorren [41] | 0.48–0.58 | 0.05–0.34 | Bedrock |
| Spadari et al. [18] | 0.43–1.85 | 0.54–0.96 | Lithic sandstones |
| He et al. [42] | 0.31 | 0.83 | Weathered limestone |

In this model, the initial direction and position of the rockfall at the top of slope slightly change, and nine rockfall trajectories are obtained. Figure 4b is the previous numerical prediction by Yan [20], and Figure 4c shows the numerical prediction with the above parameters. The proposed approach showed a more suitable for the parabolic trajectory of the field test compared with that of Yan's model, which had large energy dissipation in the second plateau, so that the trajectory shows poor agreement with the field test. The results show that the proposed approach shows good agreement with the trajectory obtained from the field test, including the scattered trajectories and sudden changes in trajectories when blocks collided with the slope.

Figure 6 shows a comparison of the predicted horizontal movement distance (Figure 6a), lateral offset (Figure 6b), and arrival time at the bottom of the slope (Figure 6c) with experimental data and previously numerical predictions [20]. For the horizontal movement distance, the rockfall rolled along the X axis, and the lateral offset was the farthest distance in which the trajectory varied from the initial position of the rock in the Y direction. Dispersion was observed in both the predicted and test results, which may have been due to the difference in the initial attitude of the column block in the numerical simulations and tests. Compared with Yan's prediction [20], the predicted average values of the

horizontal movement distance, lateral offset, and arrival time of the slope bottom of this the proposed approach were closer to the corresponding experimental values, and the numerical dispersion was less than Yan's.

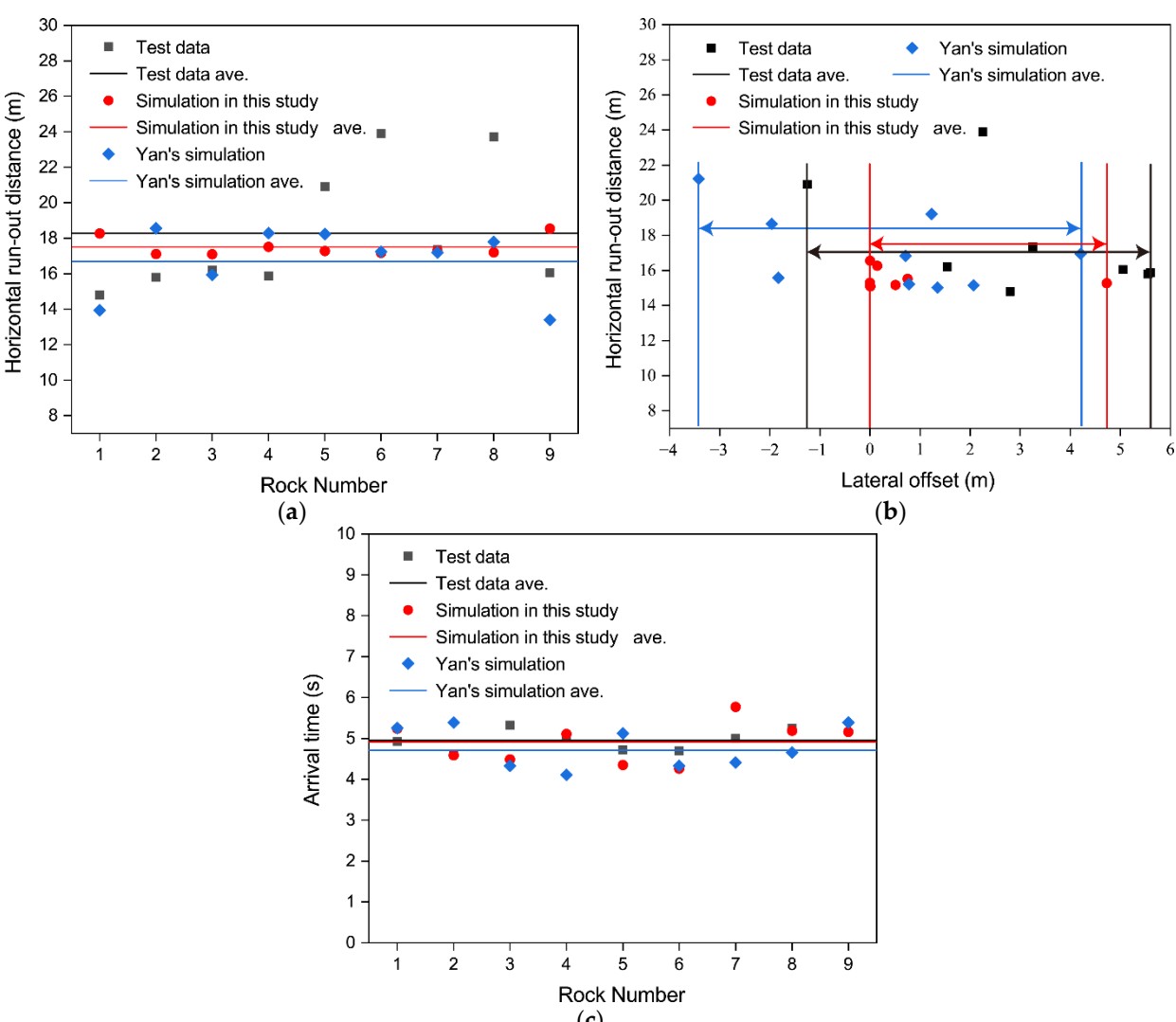

**Figure 6.** Comparison of kinematic properties between experimental data and numerical prediction. (**a**) Horizontal run-out distance; (**b**) lateral offset; (**c**) arrival time.

### 2.3. Determination of Damping Parameters for Rockfalls of Different Sizes

To further explore the relationship between the block size and damping parameters, and to ensure that the damping input parameters were reasonable, experiments were performed to determine input parameters $f_{min}$ and $\xi_{min}$ for different block sizes. Repeated experiments based on the numerical simulation procedure in Section 2.2 were conducted. The values of input parameters $f_{min}$ and $\xi_{min}$ were obtained when $R_n \approx 0.35$ for cylindrical blocks of different sizes (diameter × length) (Table 2). According to Table 2, linear regression prediction was performed among $R_t$, $V$, and $\xi_{min}$, and Equation (7) was obtained with $R^2 = 0.74$:

$$\xi_{min} = -9.71R_t + 0.12V + 6.31 \tag{7}$$

**Table 2.** COR Values of $\xi_{\min}$ for blocks of different sizes when $R_n \approx 0.35$.

| Block Size (m) | $R_n$ | $R_t$ | $f_{min}$ | $\xi_{min}$ |
|----------------|-------|-------|-----------|-------------|
| 0.14 × 0.10 | 0.350 | 0.521 | 0.015 | 0.95 |
| 0.50 × 0.50 | 0.354 | 0.614 | 0.015 | 0.44 |
| 1.00 × 1.00 | 0.356 | 0.629 | 0.015 | 0.18 |
| 1.50 × 1.50 | 0.357 | 0.672 | 0.015 | 0.12 |
| 2.00 × 2.00 | 0.345 | 0.718 | 0.015 | 0.08 |

## 3. Application of the COR Values—A Case Study

### 3.1. Study Area

Figure 7a shows the simulated slope threatened by rockfall at Lehong Tunnel in the Shouwang to Hongshan section of the Duxiang expressway in Zhaotong city, Yunnan province, China. A field investigation of the upper collapse surface was performed to determine the distribution of the dangerous rock in this area and analyze the basic characteristics of the corresponding rock blocks and the possibility of rockfall hazards.

Dangerous rocks mainly existed in Mountain Zones 1–3 above the tunnel entrance (Figure 7b). Combined with field investigations and UAV image data analysis, 12 dangerous rock areas, 2 boulder areas, and 1 rockfall accumulation area were identified. Dangerous Rock Blocks A and F in Area 1 that directly threaten the tunnel exit were selected for simulation tests.

Figure 8 shows the dangerous rocks in Area 1. Due to their high potential of causing harm to the tunnel exit, Rock Blocks A and F were analyzed (Table 3).

(1) Dangerous Rock Block A (Figure 8a) is located at the top of the left side of Mountain Zone 1 in the survey area with a slope of 70–90°. Rock A is a fractured limestone block with joints. There was a historical collapse event in this area with a partially empty wedge-shaped groove at the base of the block.

(2) Dangerous Rock Block F (Figure 8b) is located in the middle of the left side of Mountain Zone 1 in the survey area. Rock F is limestone with a block structure and joints. There are mainly two groups of joints in this dangerous rock area that bisect different rock layers and have led to rock blocks with missing bottoms.

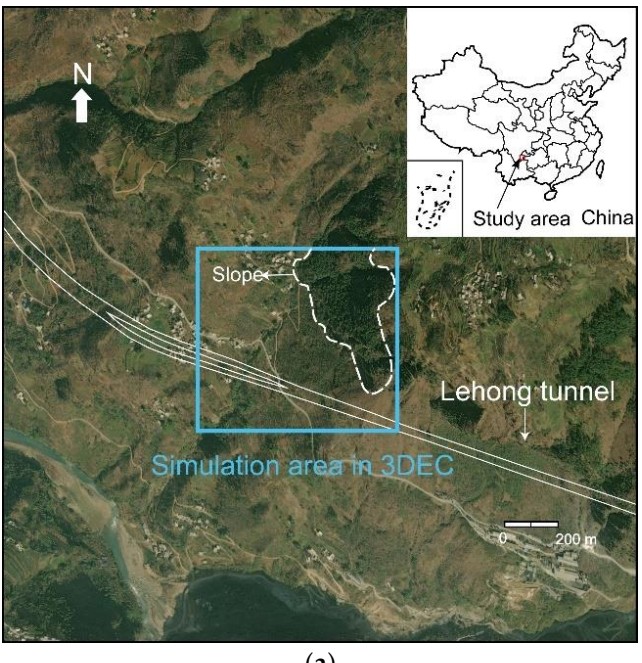

(**a**)

**Figure 7.** *Cont*.

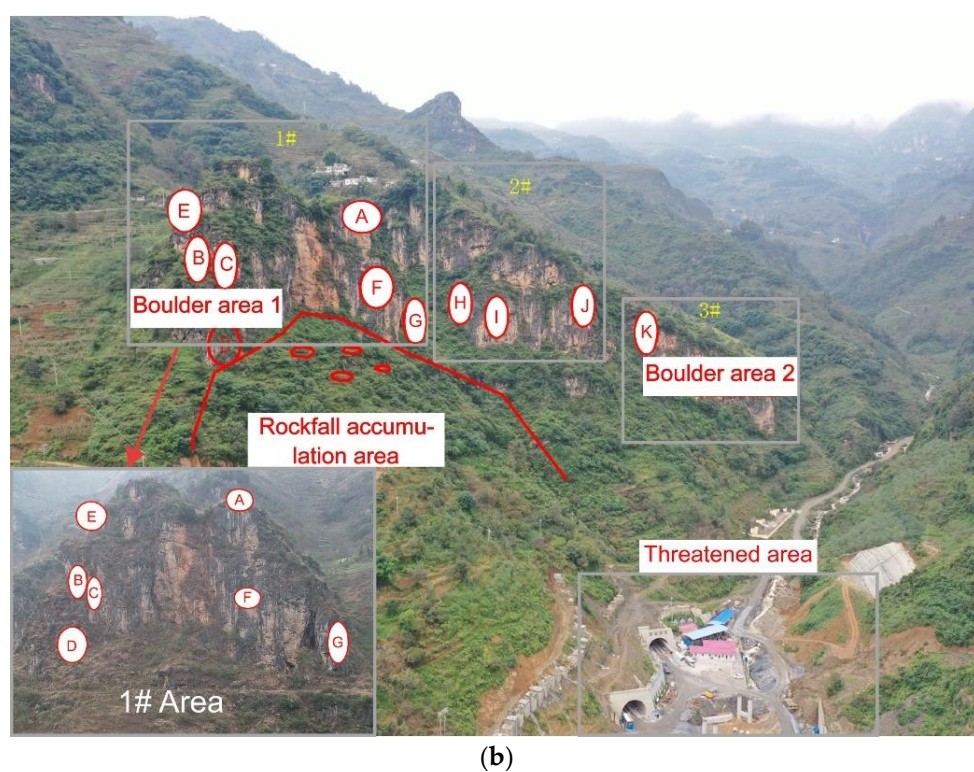

(**b**)

**Figure 7.** General situation of dangerous rocks on the upper slope of Lehong tunnel. (**a**) Satellite image of upper slope of Lehong Tunnel and (**b**) distribution of dangerous rocks in the upper part of Lehong Tunnel.

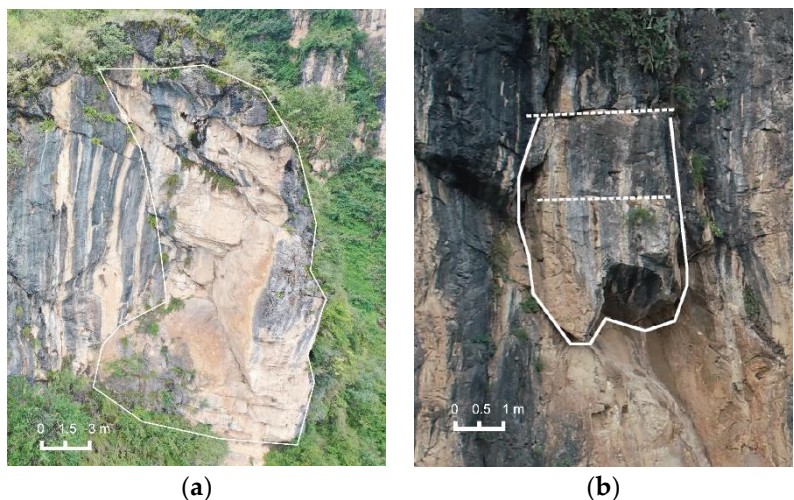

(**a**)　　　　　　　　　　　　　　　　　　　(**b**)

**Figure 8.** General situation of dangerous rock on the upper slope of Lehong Tunnel. (**a**) Rock A; (**b**) Rock F.

**Table 3.** Dangerous rock scale and movement form.

| Rock Number | Distribution Elevation (m) | Scope | | | Starting Movement Form |
|---|---|---|---|---|---|
| | | **Length (m)** | **Width (m)** | **Thickness (m)** | |
| A | 1562–1536.7 | 31.7 | 9.06 | 6.62 | Drop and pull |
| F | 1517–1510.2 | 6.6 | 3.43 | 2.9 | Drop |

*3.2. Rockfall Movement Simulation*

3.2.1. Overview

To investigate the prediction effect of the corresponding COR–damping value in the rockfall case, a numerical rockfall-slope model was built in 3DEC. As illustrated in Figure 9, the research process was divided into three main parts: the preprocessing of the numerical modeling (Steps 1–4), numerical modeling (Steps 5–8), and the analysis of the numerical results (Steps 9–11).

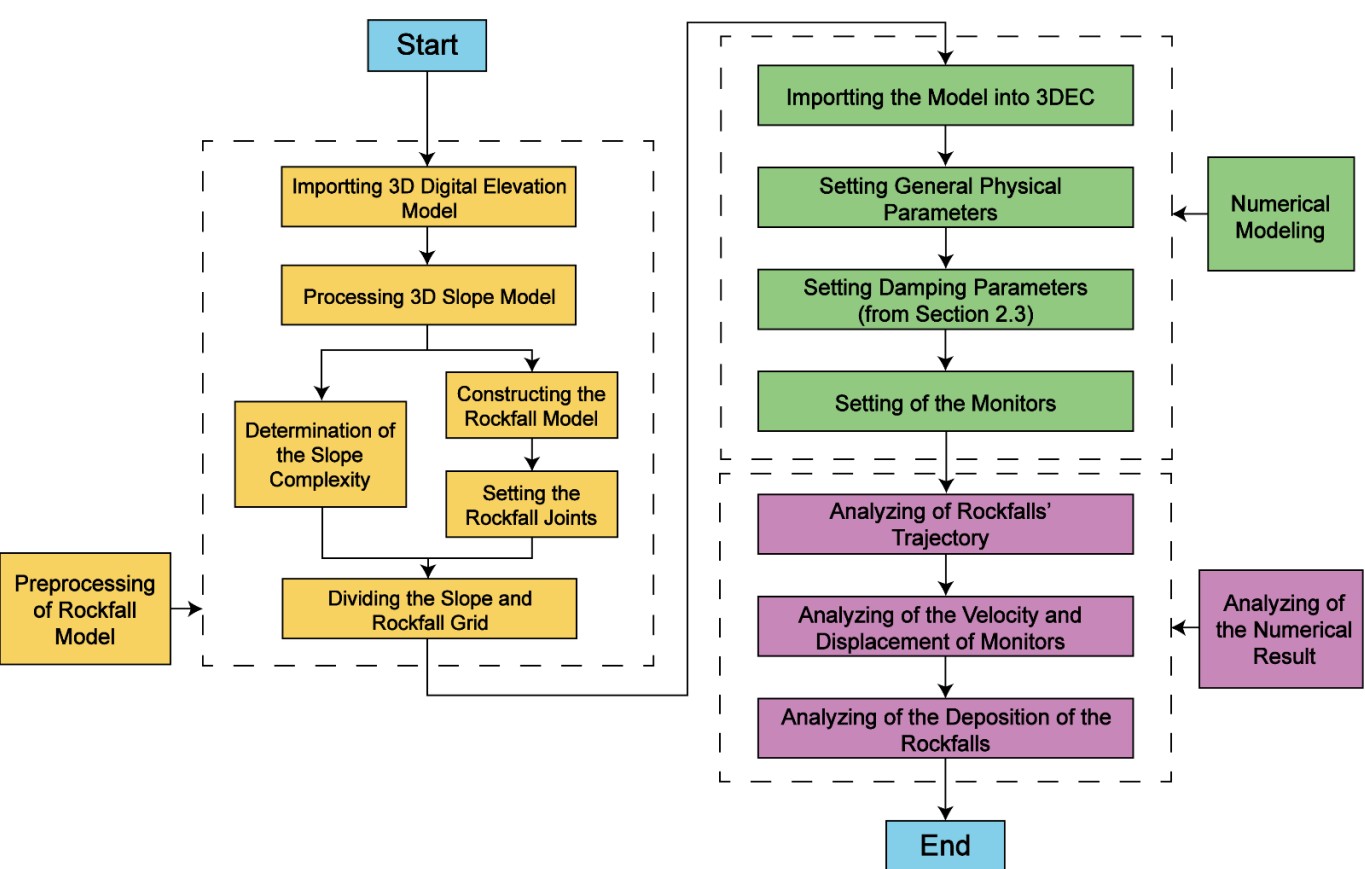

**Figure 9.** Flowchart of the numerical investigation on numerical rockfall modeling.

3.2.2. D Terrain Creation

Figure 10 shows a local 3D terrain model generated with 3D digital elevation data with 5 × 5 m resolution. The terrain grid consisted of more than 2800 tetrahedral grids. In order to improve the computational efficiency, the surface was densely divided into grids with an average side length of 30 m, and the bottom of the model was divided into grids with a side length of 500 m to reduce the number of grids. Additionally, to ensure that the rockfall model fit the slope surface, a meshing treatment was used for the contact zone between the slope and the rockfall. The source area of the rockfall is shown as a blue block, and the threatened area corresponds to the red area in Figure 10a.

The simulations of Rockfalls A and F are shown in Figure 11. According to its structure, Rock A was divided into blocks with a side length of 1.5–2.0 m; according to the structure of Rock F, one division was implemented with a division boundary near the middle of the rock.

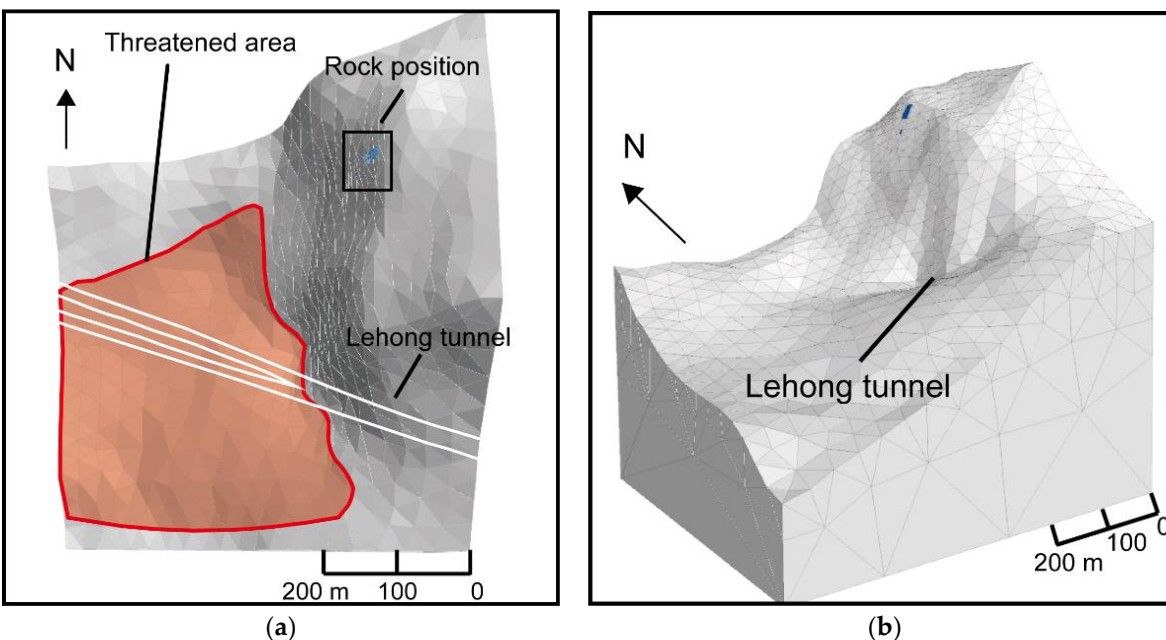

**Figure 10.** Three-dimensional terrain and dangerous rock simulation in Area 1. (**a**) Top view; (**b**) overview.

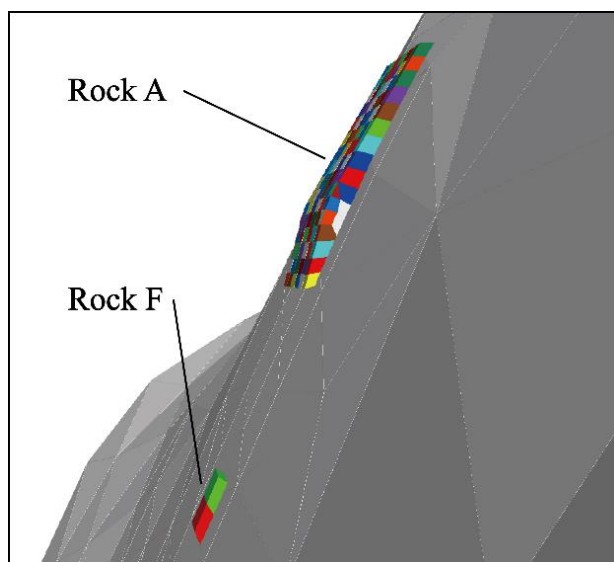

**Figure 11.** Rock simulation in Area 1.

3.2.3. Mechanics and Numerical Parameters

Table 4 lists the physical and mechanical parameters of the falling rock used in the numerical analysis. It was assumed that all test blocks were linearly elastic and that gravity was the only force driving the sliding rock downward on the slope of Lehong Tunnel.

All joints follow the Coulomb sliding joint model [35]. The cohesion, tensile strength, and dilation angle of the joints were assumed to be 0.0 kPa, 0.0 kPa, and 0.0°, respectively. The internal friction angle of the rockfall was set to 35° according to the previously discussed rockfall parameters.

The 3DEC automatically suggests a time step of $\Delta t = 6.3615 \times 10^{-7}$ s, and Rayleigh damping was set only to consider the stiffness component and reduce unnecessary numerical vibrations. To ensure that the dynamic behavior of the rockfalls remained within a reasonable range, the Rayleigh damping input parameter $f_{min} = 0.015$ verified in Section 2.3 for a side length of 1.5–2.0 m was used, and $\xi_{min}$ was interpolated to 0.10.

**Table 4.** Numerical simulation parameters.

| Item | | Value |
|---|---|---|
| Base rock | Young's modulus (GPa) | 50 |
| | Poisson's ratio | 0.3 |
| | Density (kg/m$^3$) | 2700 |
| Falling rock | Mass (kg) | 9000–21,000 |
| | Young's modulus (GPa) | 50 |
| | Poisson's ratio | 0.3 |
| | Density (kg/m$^3$) | 2630 |
| Joint | Cohesion (kPa) | 0 |
| | Friction angle (°) | 35 |
| General numerical parameters | Joint normal stiffness, $k_n$ (MPa/m) | 4.5 |
| | Joint shear stiffness, $k_s$ (MPa/m) | 3.0 |
| | Input parameter, $f_{min}$ | 0.015 |
| | Input parameter, $\xi_{min}$ | 0.10 |
| | Time step, $\Delta t$ (sec) | $6.3615 \times 10^{-7}$ |

*3.3. Rockfall Simulation Results*

3.3.1. Analysis of the Rockfall Trajectory

This model's static state of rockfalls was based on a velocity of less than 1 mm/s. Figure 12 shows the entire process of dangerous rocks moving from the top to the bottom of the slope, which takes 110 s. The rockfall movement follows a sequence from bottom to top, and each rockfall causes instability in the rock above it, thus forming a continuous rockfall. The final resting positions of all rockfalls are shown in Figure 13. Most rockfalls pass through Lehong Tunnel and the expressway during lateral movement.

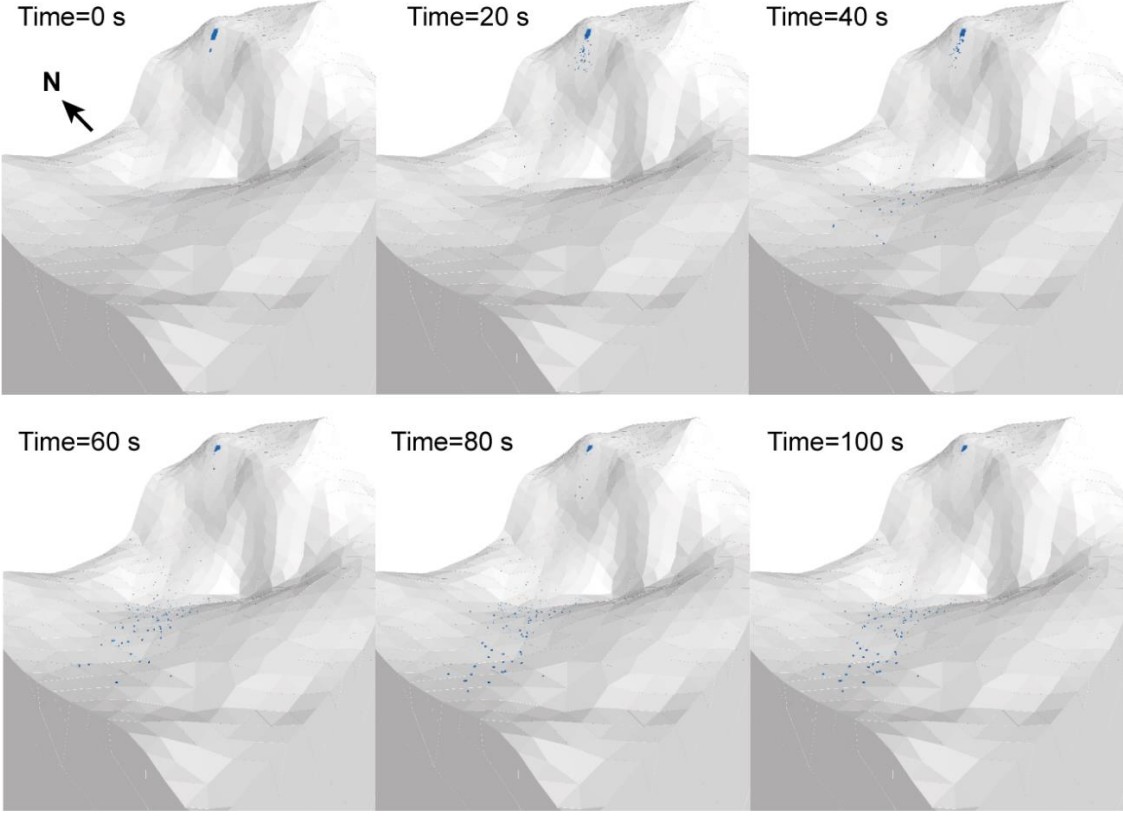

**Figure 12.** Rockfall movement. All dangerous rocks except the top of Rock A sequentially fell to the bottom of the slope.

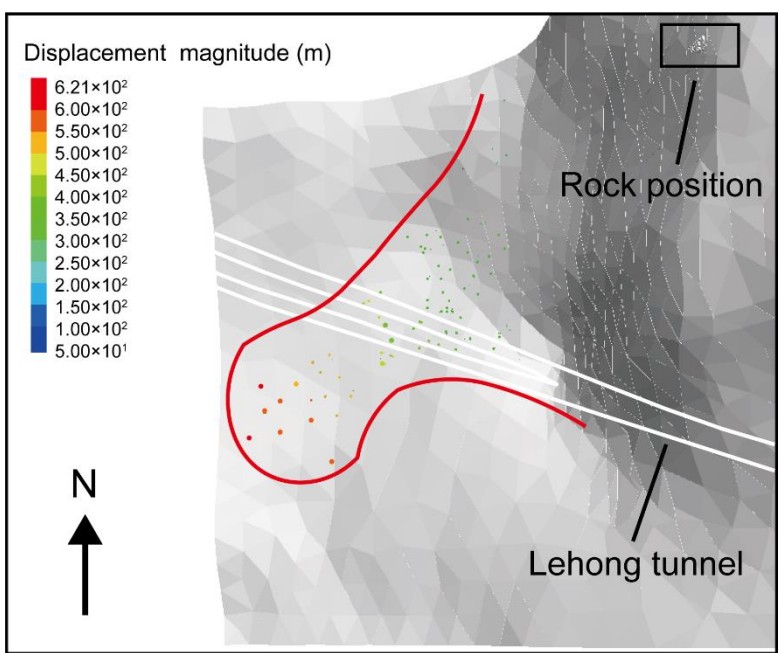

**Figure 13.** The final resting position of the rockfalls. Rockfalls are densely deposited in low-lying threatened areas. The maximal rockfall movement distance was 620 m, and two-thirds of the rockfalls moved 300–400 m.

3.3.2. Kinematic Analysis

Figure 14 shows the velocities of rockfalls at different locations. Notably, a rockfall's position significantly affects the starting time of the falling movement. In addition, some rockfalls moved suddenly at 52–63 and 103–107 s after resting at the bottom of the slope. The subsequent rockfall may hit the immobile rockfall during the rolling process, resulting in a short-term movement.

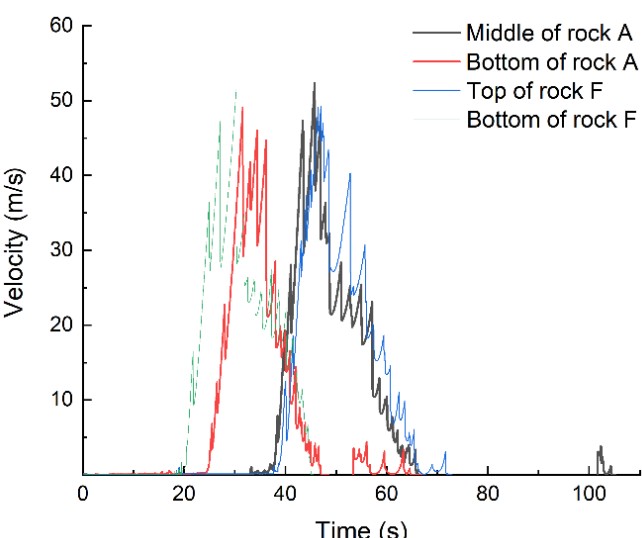

**Figure 14.** Velocity of rockfall at different locations. The maximal velocity was approximately 50 m/s, and the movement time of each rockfall was 22–29 s.

Figure 15 shows the relationship between each rockfall movement process's total and component velocities. The movement process could be divided into accelerated fall and lateral movement stages. In the accelerated fall stage, due to the slope of the study area up to 80° and less collision in movement, the rockfall speed could be close to 50 m/s. The

rockfall velocity in the Z direction contributed to most of the total velocity, but each collision between the rockfall and the slope increased the velocity in the horizontal direction. When the rockfall moved to the bottom of the slope, the sudden decrease in the slope negated the influence of the acceleration of gravity on the velocity in the Z direction for a long period. Therefore, the rockfall entered a lateral transfer movement stage dominated by the velocity in the horizontal direction. At that time, the rockfall speed could reach 20–25 m/s and pose a threat to safe transportation. Then, as the number of collisions with the ground increased, the energy gradually dissipated; eventually, the rockfall came to a rest.

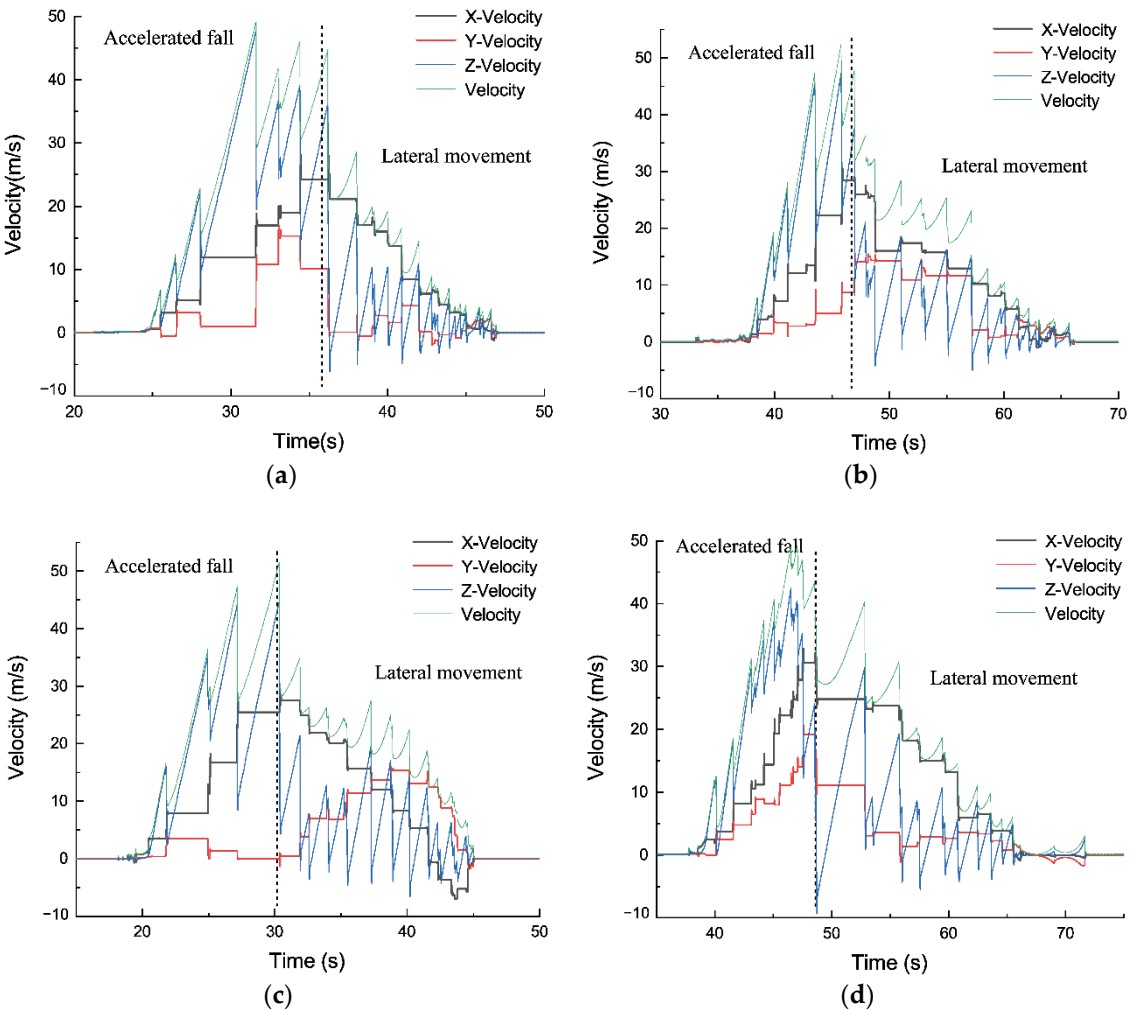

**Figure 15.** Total velocity and component velocity of rockfalls. (**a**) Bottom of rock A; (**b**) middle of rock A; (**c**) bottom of rock F; (**d**) top of rock F.

Figure 16a shows the angular velocity change during the rockfall movement process, with a maximal value of approximately 20 rad/s. The angular velocity of the rockfall increased during the accelerated fall stage and gradually decreased during the lateral movement stage, which is consistent with the change in horizontal velocity. Figure 16b shows the ratio of the kinetic energy to the rotational energy of the rockfall. The maximal kinetic energy was approximately 18 MJ, and the maximal rotational energy was approximately 1 MJ. Rotation accounted for less energy in the entire rockfall movement process and displayed a relatively consistent trend with kinetic energy changes. Therefore, most of the gravitational potential energy evolved into kinetic energy.

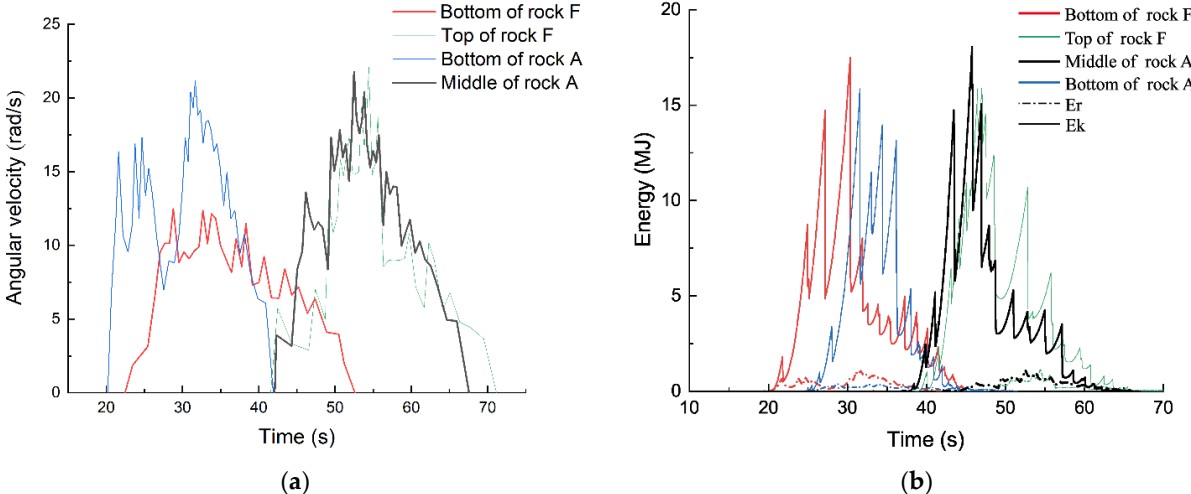

**Figure 16.** Rockfall rotation characteristics. (**a**) Angular velocity; (**b**) kinetic and rotational energy relationship.

### 3.3.3. Empirical Formula Validation of Numerical Model

Due to the lack of historical data and field tests in application cases, an empirical distance formula (Equation (8)) with a similar reach angle and volume of rockfalls was selected for verification [7] to investigate the rationality of this numerical model.

$$L = 1.1906H - 7.4717 \tag{8}$$

where $L$ is the block mass, $H$ is the angular acceleration vector (Figure 17).

This empirical formula based on the two-dimensional slope model describes the relationship between rockfall elevation $H$ and movement distance $L$. In order to apply this formula to the numerical model, three-dimensional expansion was required. Figure 18a shows the contour lines of the simulation area. In order to facilitate calculation, the figure's lower left corner was set as the origin, and the distance was marked in the X and Y directions. The distribution points of rockfall show that the rockfall movement direction presented an included angle of 25 ° with the x axis. From this, we could roughly determine the rockfall movement distance:

$$S = L / \cos 25° \tag{9}$$

In Figure 18a, 10 contour lines with an elevation of 1107–1332 and an interval of 25 m are marked with 1–10, respectively. On the basis of elevation $H_0 = 1545$ m of the starting point of rockfalls, $\Delta H$ could be obtained by subtracting the elevation of the 10 contour lines above. Then, $\Delta L$ and $\Delta S$ were calculated with Equations (8) and (9), and are summarized in Table 5.

**Table 5.** Empirical formula calculation results.

| Number | Elevation | | Distance | Displacement |
|---|---|---|---|---|
| | $H$ (m) | $\Delta H$ (m) | $\Delta L$ (m) | $\Delta S$ (m) |
| 1 | 1107 | 438 | 514.01 | 567.16 |
| 2 | 1132 | 413 | 484.25 | 534.32 |
| 3 | 1157 | 388 | 454.48 | 501.47 |
| 4 | 1182 | 363 | 424.72 | 468.63 |
| 5 | 1207 | 338 | 394.95 | 435.79 |
| 6 | 1232 | 313 | 365.19 | 402.95 |
| 7 | 1257 | 288 | 335.42 | 370.10 |
| 8 | 1282 | 263 | 305.66 | 337.26 |
| 9 | 1307 | 238 | 275.89 | 304.42 |
| 10 | 1332 | 213 | 246.13 | 271.58 |

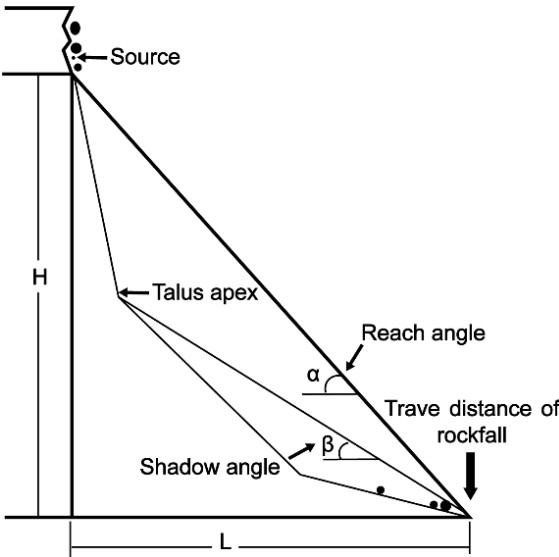

**Figure 17.** Topographic angles measured on slope profile [5].

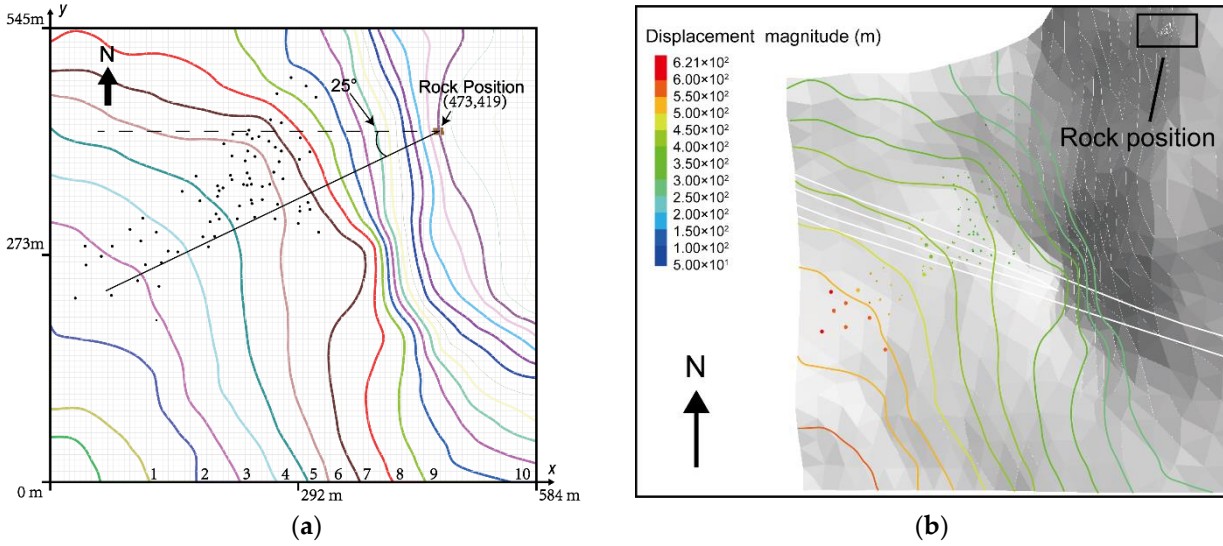

(**a**)　　　　　　　　　　　　　　　　　　　　　(**b**)

**Figure 18.** Rockfall resting position. (**a**) Contour map; (**b**) estimated distance of different contours.

In Figure 18b, 10 contour lines are colored according to the legend of displacement magnitude after the empirical formula calculates the corresponding distance S. The results show that most of the rockfall distance in the numerical simulation was consistent with the estimated distance of different contours. The larger mass and less damping may have caused the mismatch between the farthest rockfall distances and the formula results.

## 4. Discussion

The rockfall shape and the geometry of the sloping terrain had a crucial impact on the accuracy of the rockfall prediction model [11]. Although many researchers have used terrain models with a degree of complexity [11,19], they are still macroscopic predictions on a relatively large scale, and cannot accurately predict the affected area and final resting position of rockfall according to the practical terrain. Thus, we imported the DEM data of the study area into 3DEC with suitable precision. Then, the geometry of the local joints and slope terrain was explicitly used to generate polyhedrons in the source area through field investigation. The simulation results (Figure 12) show that, in the case of a large number of blocks, the movement characteristics were consistent with the previous research

results [11,19]. However, the calculation speed of 3DEC increased with the number of blocks [9], indicating that it had obvious advantages when the number of blocks was large. Moreover, the final resting position of the rockfall (Figure 13) shows that the rockfall's trajectory had a significant correlation with the terrain relief. This correlation further increased as terrain accuracy increased. Furthermore, few related papers dealt with the performance comparison of 3DEC and previously popular 3D-DDA. As far as the current research results are concerned, due to the increasing number and irregular shapes of blocks, 3DEC may be more efficient and have better application prospects for the simulation of large-scale rockfall hazards.

On the other hand, the COR, as an important parameter to describe the energy dissipation in rockfall collisions, directly affects the prediction accuracy [3]. Therefore, after considering the geological conditions (lithology, weathering degree, and other parameters) of the predicted slope, the COR value of the field test was also considered (we referred to He et al. [42] with similar geological conditions, that is, $R_n = 0.35$) through field tests [38] to verify the rationality of the trajectory prediction model at this COR value. The trajectory of the proposed approach was more consistent with the field test and more suitable than Yan's [20] numerical prediction. After the proposed approach had been verified, the experiments were carried out on rockfalls of different sizes when $R_n = 0.35$, and the corresponding values of multiple sizes and input parameters were obtained. Then, the parameter values corresponding to the rockfall size were input into the 3DEC model of the predicted slope to determine the key parameters of the energy dissipation of the rockfall movement. Previously, the accuracy of simulation results was often verified by referring to historical disaster ranges [8–10] or field tests [11,19]. Therefore, this work can provide a new method for the approximate prediction of rockfall hazards. That is, in the absence of historical rockfall events or field tests on a specific slope, a reasonable COR is adopted to describe the energy consumption in rockfall collisions. The rockfall movement trajectories on complex geometric slopes can be simulated from this.

There are also some limitations in this paper that need further research. First, COR was mainly selected on the basis of experience, and simulations with multiple CORs were not performed. While the simulation results still provide useful reference for research, it would be preferable if more simulation data were available for comparison. Second, the factors affecting the selection of the COR value were not considered comprehensively. Most of the mechanical rockfall parameters in this proposed approach were empirical values, and it is necessary to conduct a sensitivity study between the mechanical properties and the COR value of the rockfall. At the same time, the influence of earthquakes, rain, and vegetation on the COR value should also be considered to obtain a more comprehensive COR value evaluation system. Lastly, while actual control measures are being developed, the results of field tests and numerical simulations should be comprehensively considered. The results of the proposed approach can provide a reference for the preliminary formulation of disaster prevention measures. However, a more accurate disaster range may require further field tests and historical disaster ranges of slopes with similar geological conditions as a reference.

## 5. Conclusions

In summary, a new method for predicting rockfall trajectories was proposed in this study, and the following conclusions were obtained:

(1) A series of values were obtained by investigating the relationship between damping and COR, and the rationality of the corresponding value of damping–COR was verified with field tests and its previous numerical predictions.

(2) The corresponding damping–COR value was adopted in the simulation of typical rockfall. The results show that the maximal velocity of rockfalls was approximately 50 m/s, and the maximal kinetic energy was approximately 18 MJ. Most of the rockfalls passed the highway or the tunnel entrance and collided with it, posing a serious threat to safe transportation. In addition, some scattered blocks remained at

the top of the slope instead of falling; therefore, they should be removed to avoid secondary rockfall hazards in the future.

**Author Contributions:** Conceptualization, Y.A. and H.L.; Funding acquisition, Y.A. and H.L.; Writing—original draft, Y.A., H.L. and Z.G.; Writing—review and editing, Y.A., H.L. and Z.G. All authors have read and agreed to the published version of the manuscript.

**Funding:** This research is funded by the National Key Research and Development Plan of China (Grant No. 2019YFC1509701) supported by the Ministry of Science and Technology of China, and Hami Science and Technology Plan of China (grant No. hm2021kj08) supported by Hami Municipal Bureau of Science and Technology.

**Institutional Review Board Statement:** Not applicable.

**Informed Consent Statement:** Not applicable.

**Data Availability Statement:** The data presented in this study are available on request from the corresponding author.

**Acknowledgments:** Thanks to the National Key R&D Program of the Ministry of Science and Technology of China, Research Group on Landslide and Collapse Disaster Prevention and Control in Strong Earthquake Areas, for the relevant information provided, especially the field survey data provided by the research team of Luqing Zhang and Xueliang Wang, Institute of Geology and Geophysics, Chinese Academy of Sciences.

**Conflicts of Interest:** The authors declare no conflict of interest.

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
