# Peer review of "Numerical Investigation on the Correspondence between the Damping and Coefficient of Restitution (COR) in Rockfall Movement"

_applsci, doi:10.3390/app122010388_

Round 1

Reviewer 1 Report

The authors of the manuscript titled "Numerical Investigation on the Correspondence between the Damping and Coefficient of Restitution (COR) in Rockfall Movement" studied the rockfall problem thanks to a discrete element method tridimensional code. In order to achieve the necessary results for rockfall engineering, the authors need to perform a tuning of the parameters to simulate energy dissipation when the blocks bounce.

The study is interesting and opens new possibilities for the analysis of such systems. Anyway, there are several issues that raise and that must be addressed:ç

1. A detailed description of the various propagation models must be included (considering hybrid, lumped mass and rigid body analyses). Non-Chinese would be appreciated.

2. Can the authors include a simplified model for studying the correlation between restitution coefficients and damping? It seems that the a real analytical model connecting the two quantities is missing. Can the authors delineate such a model to address the problem?

3. The validity of the proposed approach must be discussed. As in any rockfall simulation, an initial back-analysis of a real event must be performed to "tune" slope parameters. How the results can be extended to other slopes?

4. An operative flowchart helping experts and technicians in modelling rockfalls thanks to DEM can be interesting and helpful.

5. A review of the format, lettering and margins is required (eg., p.8 bottom)

6. Figure 11 is difficult to understand. Can the authors improve its readability?

Reviewer 2 Report

Dear authors,

thank you for this contribution. I also think that it is important to investigate modelling details more closely and I think, your approch is acceptable.

The first sentence of the abstract could be more specific rather than the usual mentioning of rockfall hazard. E.g. something like "The rockfall process is characterized by bounces of the block on the ground."

L27: remove "houses," and "school"

L25-47: Please, be aware that the usual usage of the COR is a try to get the block jump in the simulation. I.e., it is a model only. Have try and let a rock drop onto the ground. It barely jumps. The jumps of the trajectory in the field originate only from the irregular shape of the block in combination with its rotation. Maybe, you can add something like this?

L48-58: to which category do you sort the model presented in https://ramms.slf.ch/en/modules/rockfall.html

L110: "acceleration" --> "velocity"

L124ff: you could mention the difference between global and contact damping

Figure 4: It is not visible how far the simulated trajectories go to the right. Would it be possible move the terrain a little bit downward in the graphs to make the trajectories better visible?

L183: the equations 4 show a COR based on velocity change, not energy. Change "energy" to "velocity". Please, be aware that different COR exist for energy and velocity changes (see for example Heidenreich, B. (2004). Small-and half-scale experimental studies of rockfall impacts on sandy slopes (No. THESIS). EPFL.)

L204: remove the last "the"

L211-221: add once the reference number of "Yan" in this paragraph as well.

L237: Is this really a verification? You did a case study with you model but I miss a comparison with real field data or at least with one or two other trajectory simulation models.

Page 8: figure 8 has a caption only. Please, add the content.

Page 9: You have to correct the figure numbering. Currently, two caption state "Figure 8".

Table 4 has to be improved:

- The block mass is missing

- put the units into own columns

- It is not clear from which line the parameters for the base rock, the falling rock and the joint start and end. Please, improve the structure, visibility

- The "remarks" column seems to be not necessary

- put the two stiffenesses into two lines

Figure 11: Sorry, but I can barely see any difference for all twelve pictures. You have to add more contrast to the trajectories, make them thicker or the pictures larger, etc.

Figure 13: I have the feeling that 50m/s is rather fast. Can you confirm this value from field tests or other rockfall simulation codes?

Reviewer 3 Report

The paper is devoted to the numerical simulation of rockfall movement using 3DEC computer code based on the discrete element method. As I understood the novelty of the paper is the approach to the choice of parameters that describe the damping effects. Authors proposed to find by trial and error in direct numerical simulations such damping parameters that would provide reasonable value of restitution coefficient. The proposed approach was illustrated in the specific example. Damping parameters calibrated using full-scale experiments [37] were used for the numerical analysis of the rockfall at Lehong Tunnel in China.

I think the obtained results are interesting and useful for the community. The paper can be published in the Applied Sciences after the following comments will be addressed:

1. I think Section 2.1 is not written very well. In fact it is almost completely taken from [33]. Mass moment of inertia in (1b) is not a vector, it should be a scalar. The text gives a poor explanation of the main principles of 3DEC code, even the basics of the mathematical model implemented in the code. The sense of parameters f and xi are not explained. It could be done at least as in 3DEC manual, http://docs.itascacg.com/3dec700/3dec/docproject/source/options/dynamic/dynamic_damping.html?node3404.

2. On pages 6 and 14 authors use the phrase a “proposed model” when they talk about their findings. It is not correct. They don’t propose any new model either in the sense of a defining system of equations or in the sense of some new statement of the problem. In fact they just fit some parameters of the model for the simulation results to be more reasonable. I suggest using the phrase “proposed approach” or something like that.

3. A proofreading of the paper is necessary. For example:

- The sentence “Due to the complex contact transformations…” is without a subject and a predicate.

- Authors should pay attention to the consistency of the notations throughout the text. In the Section 2.1 small letter k is used for stiffness while in Section 2.2 the capital K is used for the same purpose.

- The phrase “and the friction angle of the block was set…” at the end of the third paragraph of the Section 2.2 should be placed to some correct place.

- The sentence “In 3DEC manual states…” should be re-written.

Round 2

Reviewer 1 Report

The authors have addressed all the points raised, expect one. Can they provide a detailed description of the various propagation models available in the literature (considering hybrid, lumped mass and rigid body analyses). The reviewer is referring to the propagation of the phenomenon, not the damping in the model considered by the authors.

Reviewer 2 Report

all my previous comments have been handled sufficiently.

Author Response

Dear reviewer:

Thanks very much for your kind work and consideration on publication of our paper. On behalf of my co-authors, we would like to express our great appreciation to you.

Thank you and best regards.

Yours sincerely,

Hongyan Liu